# An Improved Co-Training and Generative Adversarial Network (Diff-CoGAN) for Semi-Supervised Medical Image Segmentation

Guoqin Li [1,2], Nursuriati Jamil [2,*] and Raseeda Hamzah [3]

1  Taiyuan Institute of Technology, Taiyuan 030008, China
2  College of Computing, Informatics and Media, Universiti Teknologi MARA, Shah Alam 40450, Selangor, Malaysia
3  College of Computing, Informatics and Media, Universiti Teknologi MARA, Melaka Branch, Merlimau 77300, Melaka, Malaysia
*  Correspondence: lizajamil@computer.org

**Abstract:** Semi-supervised learning is a technique that utilizes a limited set of labeled data and a large amount of unlabeled data to overcome the challenges of obtaining a perfect dataset in deep learning, especially in medical image segmentation. The accuracy of the predicted labels for the unlabeled data is a critical factor that affects the training performance, thus reducing the accuracy of segmentation. To address this issue, a semi-supervised learning method based on the Diff-CoGAN framework was proposed, which incorporates co-training and generative adversarial network (GAN) strategies. The proposed Diff-CoGAN framework employs two generators and one discriminator. The generators work together by providing mutual information guidance to produce predicted maps that are more accurate and closer to the ground truth. To further improve segmentation accuracy, the predicted maps are subjected to an intersection operation to identify a high-confidence region of interest, which reduces boundary segmentation errors. The predicted maps are then fed into the discriminator, and the iterative process of adversarial training enhances the generators' ability to generate more precise maps, while also improving the discriminator's ability to distinguish between the predicted maps and the ground truth. This study conducted experiments on the Hippocampus and Spleen images from the Medical Segmentation Decathlon (MSD) dataset using three semi-supervised methods: co-training, semi-GAN, and Diff-CoGAN. The experimental results demonstrated that the proposed Diff-CoGAN approach significantly enhanced segmentation accuracy compared to the other two methods by benefiting on the mutual guidance of the two generators and the adversarial training between the generators and discriminator. The introduction of the intersection operation prior to the discriminator also further reduced boundary segmentation errors.

**Keywords:** semi-supervised learning; medical image segmentation; co-training; GAN; Diff-CoGAN

## 1. Introduction

Semi-supervised learning is favored when labeled data are scarce or expensive to obtain, but a large amount of unlabeled data are available, such as in the case of medical image segmentation [1–5]. When performing image segmentation using semi-supervised learning, the accuracy of predicted map is crucial as it determines the predicted labels for the unlabeled data points in the dataset. For example, the authors in [1] combined post-processing with self-training to train a segmentation network with labeled data, then unlabeled data were fed into the network to get the predicted map. Then, the predicted map was optimized by post-processing. The refined map was directly taken as additional ground truth for updating the network parameters. In [2], the authors proposed a two-stream network and a hybrid method based on model distillation and data distillation to generate a predicted map for unlabeled data, which was then fed into the network for training. Even though additional operations, such as post-processing and hybridization, were used,

predicting labels of an unlabeled dataset for a training model remains a challenge. Due to the lack of ground truth, it is difficult to generate a high-confidence predicted map. The generation of a high-quality predicted map is vital in semi-supervised learning as it affects the performance of training a segmentation model.

Co-training of semi-supervised models introduced by Blum and Mitchell [6] is also beneficial for image segmentation as the predicted map generated by one model can be used by another model to improve its own prediction. Co-training firstly trains multiple models by using labeled data, in which each model is trained sufficiently with one data view (e.g., data source and rotation). Then, the predicted maps generated from different views should be agreed upon for the same unlabeled data, which enforces multiple models to be generalized well to unlabeled data [7–17]. For medical image segmentation, the authors in [7] proposed an uncertainty-aware multi-view co-training framework to segment 3D medical images in which an uncertainty-weighted label fusion mechanism was proposed to estimate the reliability of a predicted map for each view. In [8], the authors proposed a deep adversarial co-training method for 2D medical image semantic segmentation. The authors used multiple models to generate predicted maps and then fused them to get the mean results. At the same time, adversarial samples were introduced to capture the difference between the models, thereby enabling the models to learn more complementary information in the training process. Although the fusion of predicted maps can improve training performance, the measurement to accurately estimate the confidence of predicted maps is still an essential problem [17]. During fusion, several predicted maps with low segmentation performance will affect the training outcome. Therefore, improving the accuracy of generated predicted maps is one of the aims of our proposed research.

Generative adversarial network (GAN) is a deep learning algorithm comprising a generator network and a discriminator. GAN uses adversarial training [18] between the generator and the discriminator and has shown that the generator can influence the generation of a reliably predicted map on unlabeled data to fool the discriminator. With adversarial training, the discriminator's ability to distinguish true and fake data is improved, and it promotes the generator to produce higher-quality fake data that are closer to the true data. Consequently, the overall performance of GAN is improved. In [19], GAN was proposed to generate a dataset of synthetic embryo images by training on real human embryo images. The aim was to perform data augmentation for future work of classification, analysis, and training. The proposed model achieved the highest quality of synthetic images for single-cell embryo images. GAN has also been used in medical image segmentation in recent years [20–26]. For example, the authors in [20] proposed a deep adversarial network (DAN) to achieve the medical image segmentation task. The authors designed an adversarial loss function, which better controlled the training process using unlabeled data. In [21], the authors further optimized the discriminator to output confidence maps of predicted maps. Only predicted maps with high confidence values participated in the model training. In [26], the authors fine-tuned a pix2pix-GAN model, and then a trained GAN model was used to automatically segment lung infection from CT images. All the abovementioned work used GAN consisting of one generator corresponding to one discriminator to achieve medical image segmentation. There have also been many medical image segmentation approaches based on the extension of GAN, as described in the paper [27]. All except CycleGAN [28] contained one generator and one discriminator in their proposed extended GANs. CycleGAN comprised two sets of generator–discriminator in which one generator corresponded to one discriminator, meaning the adversarial training was between one generator and one discriminator. Meanwhile, the adversarial training in our proposed Diff-CoGAN is between two generators and one discriminator, which further facilitates the three networks' feature extraction ability by mutual influence.

The strengths of adversarial training of GAN and co-training strategy are complementary and are the aim of this paper. Thus, the combination of a co-training strategy and GAN was proposed and named Diff-CoGAN to produce high-quality predicted maps for medical image segmentation. In Diff-CoGAN, two different generators and one discriminator

were adopted and compared to the classic and extended GANs as reviewed in [27]. The two different generators utilize the co-training strategy to provide mutual segmentation information. Furthermore, the adversarial training between the two generators and the discriminator allows better interaction among them. In [8], one generator corresponds to one discriminator in each stream of co-training, thus the mutual guidance is limited. The adversarial training in Diff-CoGAN is between two generators and one discriminator, which further facilitates the three networks' feature extraction ability by mutual influence. Thus, the generators could learn more features extracted from each other under the guidance of one discriminator, which consequently improves the quality of the predicted maps. Furthermore, Diff-CoGAN can save computing space by using one discriminator when compared to the work by [8] in which two discriminators were adopted.

The major contributions of this paper are concluded as follows:

(1) In Diff-CoGAN, there are two different generators and one discriminator (See Figures 1 and 2), in which the generators generate predicted maps for medical images, and the discriminator is used to discriminate between the ground truth and the predicted maps.

(2) In Diff-CoGAN, the two generators provide mutual segmentation information, which also supervises each other. In this paper, we introduced the intersection of two predicted maps with high confidence region produced by the outputs of the two generators as the input to the discriminator (see Section 2.3.3). Diff-CoGAN achieves higher segmentation accuracy through adversarial training because the discriminator can guide the generators to generate more accurate predicted maps, and the mutual information guidance of the two generators can also promote the improvement of segmentation performance.

(3) The two generators adopt different networks (see Sections 2.3.1 and 2.3.2) to increase the diversity of extracted features, consequently providing complementary information for the same data along with the strategy of co-training.

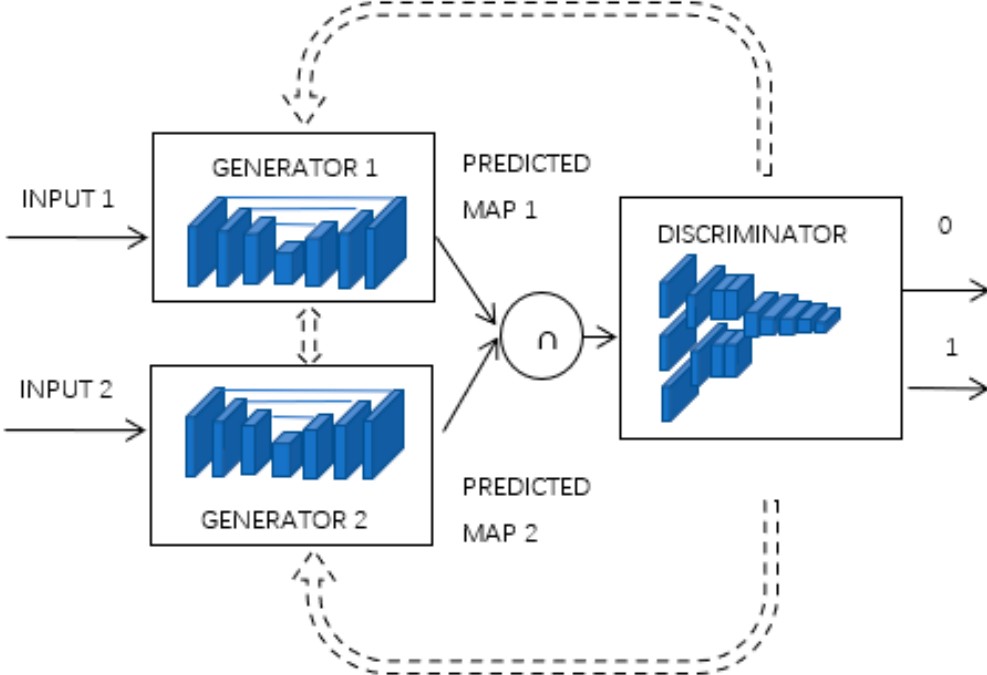

**Figure 1.** The brief framework of Diff-CoGAN comprises two generators and one discriminator.

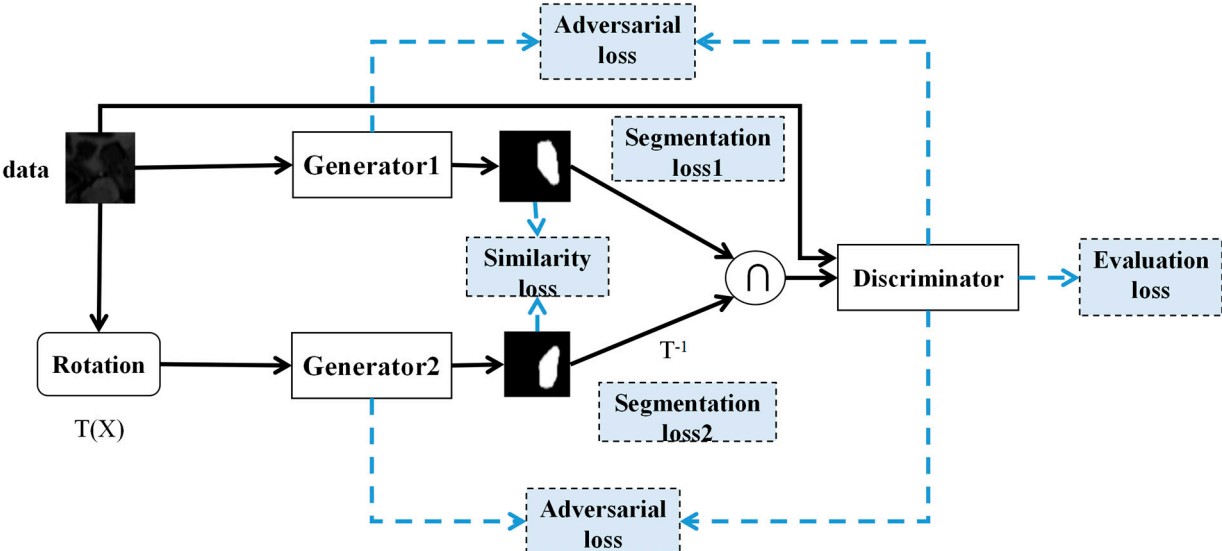

**Figure 2.** Configurations of Diff-CoGAN framework showing two generators and one discriminator. The input data comprise 2D labeled and unlabeled image datasets. The input to Generator 1 is the original dataset, while Generator 2 is fed the transformed (TX) dataset. T(X) is a 180° rotation of the original dataset. Consequently, predicted maps are generated by the two generators and are intersected (notated by the intersection, ∩) to be the input to the discriminator. Prior to intersection, the predicted map output from Generator 2 is inversed, which is notated by $T^{-1}$.

## 2. Materials and Methods

This section describes our proposed Diff-CoGAN in detail. Firstly, the overview of the Diff-CoGAN framework is explained in Section 2.1. The networks' configuration details of Diff-CoGAN are described in Section 2.2. Finally, the training strategy of Diff-CoGAN is introduced in Section 2.3.

### 2.1. Overview of Diff-CoGAN Framework

Figure 1 shows the overall framework of Diff-CoGAN. There are two different generators and one discriminator in Diff-CoGAN. For each generator, it accepts one view of data to achieve segmentation. The data view for Generator 1 is the original image. The data view for Generator 2 is the rotated original image, and the rotation angle is 180°. The aim of the two generators is to achieve correct segmentation for the region of interest (ROI), and the output of each generator is a predicted map. The aim of the discriminator is to evaluate the quality of the predicted maps.

In classic GAN, one generator commonly corresponds to one discriminator. The generator is mainly expected to achieve segmentation (predicted map) with high similarity to the ground truth. The discriminator is expected to distinguish between the predicted map and the ground truth. The output of the discriminator is an evaluation score, ranging from 0 to 1. The predicted map has a higher similarity to the ground truth if the evaluation score is closer to 1. A score closer to 0 indicates a fake predicted map that is less similar to the ground truth. During the adversarial training process between the generator and discriminator, the performance of the generator and the discriminator is both improved by iterative optimization. The best training performance is when the generator generates a predicted map that is closest to the ground truth and can fool the discriminator, making it fail to distinguish between the predicted map and the ground truth.

Diff-CoGAN uses two generators to generate predicted maps and one discriminator to achieve quality evaluation for the generated predicted maps. Besides the basic functions in classic GAN, there are other aims of the generators and the discriminator in Diff-CoGAN. Firstly, the two generators provide mutual guidance to each other. The generators have different structures and accept different views of the data to realize segmentation. Thus,

the predicted maps from the two generators have differences, containing complementary information. During the training process, the predicted map from each generator can be taken as a fake ground truth to the other generator to learn complementary information. Then, the two generators reach a consensus for the same data segmentation through iterative optimization. Secondly, the adversarial training between the two generators and the discriminator allows them to interact with each other and improves their own feature extraction ability by mutual influence. The final difference between Diff-CoGAN and classic GAN is the input to the discriminator. In Diff-CoGAN, the intersection of the two predicted map outputs from the two generators is the input to the discriminator. The two generators both agree that the intersection region has high confidence of being the ROI. Thus, this can alleviate the effect of wrong evaluation for the discriminator.

### 2.2. Configurations of Diff-CoGAN Framework

Figure 2 illustrates the configuration of Diff-CoGAN framework and how the labeled and unlabeled data are used by the generators and discriminator to conduct image segmentation. The segmentation performances of the generators and the discriminator are measured using loss values, which are the segmentation loss, similarity loss, adversarial loss, and evaluation loss. Segmentation loss 1 and segmentation loss 2 are used to measure the segmentation performance of Generator 1 and Generator 2 by using fully supervised learning (training models using labeled data only) during the training of the Diff-CoGAN framework. A smaller segmentation loss value indicates a better segmentation performance. The similarity loss is used to evaluate the similarity between the predicted maps from the two generators. Similar predicted maps generate smaller similarity loss values. A smaller similarity loss means the two generators can reach an agreement for the input data. The adversarial loss is used to estimate the predicted map from either the labeled data or unlabeled data to appear like the ground truth. Because Diff-CoGAN only has one discriminator, the performance of the discriminator can guide the optimization of the two generators at the same time. Generator 1 and Generator 2 can influence each other through an adversarial training process and mutual guidance from similarity loss. Finally, the evaluation loss is used to measure the performance of the discriminator.

Medical image segmentation using deep learning models has commonly adopted an encoder–decoder structure, in which the encoder part is used to extract low-level and high-level feature maps and the decoder is used to recover the information of the image [29,30]. A typical model is Unet [31–33]. Meanwhile, feature maps of the same sizes between the encoder and decoder are concatenated together to compensate for the loss of information in the down-sampling steps. Even though the two generators in Diff-CoGAN use different networks to achieve segmentation, their basic network structures both contain encoder–decoder and concatenation between feature maps of the same sizes. Diff-CoGAN strategizes by using different network designs for the two generators to extract different features of the same data.

### 2.3. The Network Design of Diff-CoGAN

As stated earlier, the generators in Diff-CoGAN use different network designs with different purposes. One generator is to build a new network, and the other is to leverage the trained models by transfer learning.

### 2.3.1. The Network of Generator 1

The encoder–decoder network structure of Generator 1 is shown in Figure 3. In the encoder, the input data are first convolved with a $1 \times 1$ filter, followed by four block processing to extract feature maps of different sizes. Each block is a dense structure in which high-level features can be generated by incorporating low-level features, thus facilitating the identification of the target region [34]. In each block, all the outputs of subsequent layers are concatenated in a feed-forward manner. Meanwhile, each convolution layer (Conv) in the dense block is followed by a Batch Normalization layer (BN)

and leaky rectified linear unit (Leaky-Relu). In the encoder, four dense blocks are used to alleviate the vanishing gradient problem in the deep learning model by combining high-level and low-level features. In each block, the feature maps are subjected to a series of convolution + Batch Normalization + Leaky-Relu. Each feature map is concatenated with the previous convolved maps, and a final convolution is performed to produce a convolved output. The output of each block is further maxpooled to the next block and concatenated to the decoder. In the decoder, three upsampling steps are performed to the feature maps, and each upsampling is followed by one convolution. Concurrently, feature maps of the same size between the encoder and decoder are concatenated before convolution processing in the decoder part. Finally, three convolutions are used to gradually transit the channel of feature maps to obtain the output.

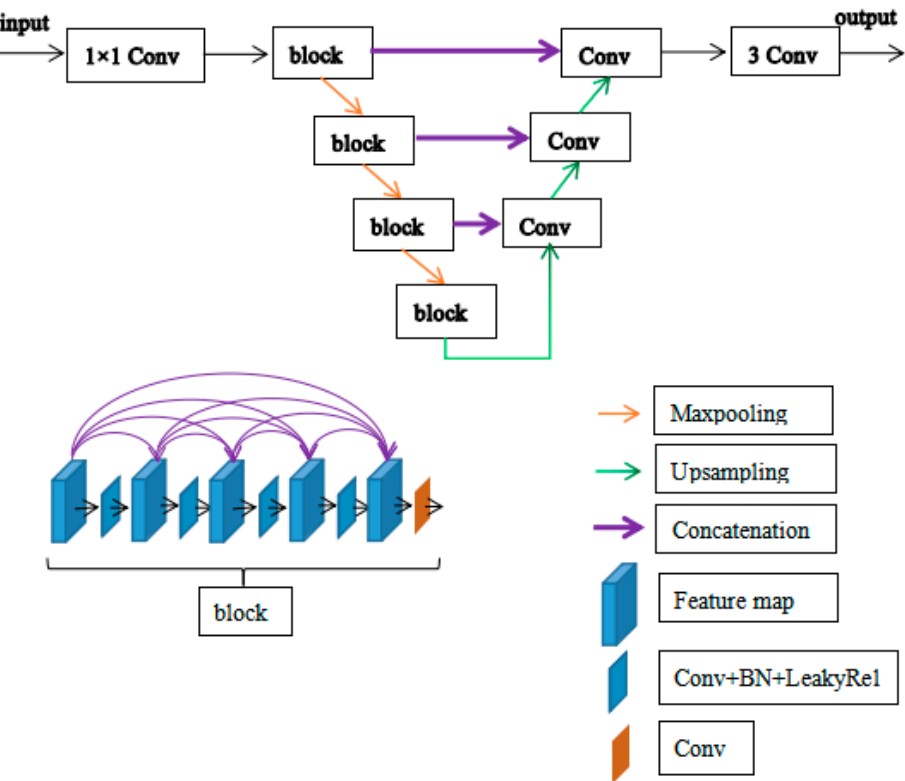

**Figure 3.** The network design of Generator 1 consists of an encoder and a decoder. The encoder extracts features using 4 dense blocks of convolution, batch normalization, and Leaky-Relu. The feature maps are then upsampled by the decoder to produce the output.

2.3.2. The Network of Generator 2

There are many mature models that have been trained for classification tasks using the large-scale dataset from ImageNet, such as VGG [35], Resnet [36], and DenseNet [37]. Therefore, these trained models can be utilized to achieve segmentation tasks by transfer learning. For medical image segmentation, the encoder can adopt some layers of the trained models to extract features by transfer learning, and the decoder is designed to recover the image information. Table 1 describes the segmentation performance of three trained models by using the Hippocampus dataset under fully supervised learning. As can be seen, the VGG16 model has the highest Dice value [38] with a minimum number of layers. Thus, the VGG16 model was adopted as the basic network of Generator 2.

**Table 1.** VGG16 shows better segmentation performance compared to ResNet50 and Dense121.

| Learning Model | Dice Value | Number of Layers |
|---|---|---|
| VGG16 | 0.741 | 16 |
| ResNet50 | 0.726 | 50 |
| Dense121 | 0.690 | 121 |

The network design of Generator 2 is shown in Figure 4. The dashed box is the trained model using VGG16 transfer learning. The output of VGG16 is upsampled to the decoder. In the decoder, the four upsampling steps are performed to the feature maps, and each upsampling is followed by one convolution. Concurrently, feature maps of the same size between the encoder and decoder are concatenated before convolution processing in the decoder. Finally, two convolutions are used to gradually transmit the channel of feature maps to produce the output.

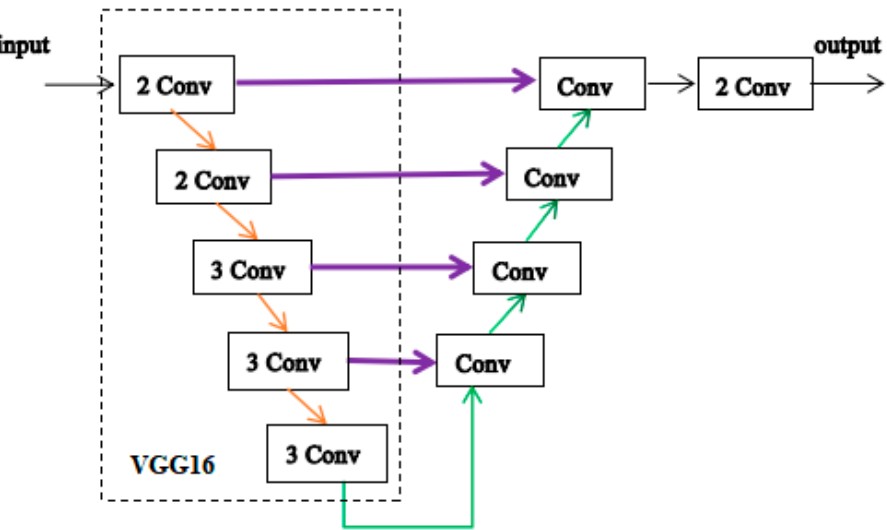

**Figure 4.** The network design of Generator 2. In Generator 2, a transfer learning model is implemented as the encoder to extract features. The decoder performs four upsampling steps, followed by a series of convolutions to produce the output.

2.3.3. The Network of the Discriminator

The discriminator network is shown in Figure 5. The input data are the segmented region of interest (ROI) and its corresponding original image. In classic GAN, the ROI for the discriminator is the predicted maps, while in Diff-CoGAN, the segmented ROI sources are from the intersection of the predicted maps generated by the two generators and the ground truth annotated by experts. The intersection part indicates that both generators agree the region is to be from a part of a target region, which means the intersection part has a high confidence of being the ROI. At the same time, the boundary errors of the predicted maps are reduced by using the intersection part in the training process. Further discussion is detailed in Section 3.4. The segmented ROI and the original image are subjected to $1 \times 1$ convolution and $3 \times 3$ convolution, respectively, to acquire the feature maps. Then, the feature maps are multiplied to fuse the information of the segmented ROI and the original images. The multiplied output is further applied with (Convolution + Batchnormalization + LeakyRelu) operation four times. Next, $3 \times 3$ convolution and $1 \times 1$ convolution are used to gradually reduce the channels of the feature map. Finally, Flatten and Dense steps are carried out to get the evaluation score. The evaluation score ranges from 0 to 1. The input data of the discriminator from the ground truth are represented with 1, while 0 represents the input data of the discriminator from the predicted map. If the evaluation score is closer to 1, this means the input data are closer

to the ground truth. In contrast, the input data closely resemble the predicted map if the evaluation score is nearer to 0.

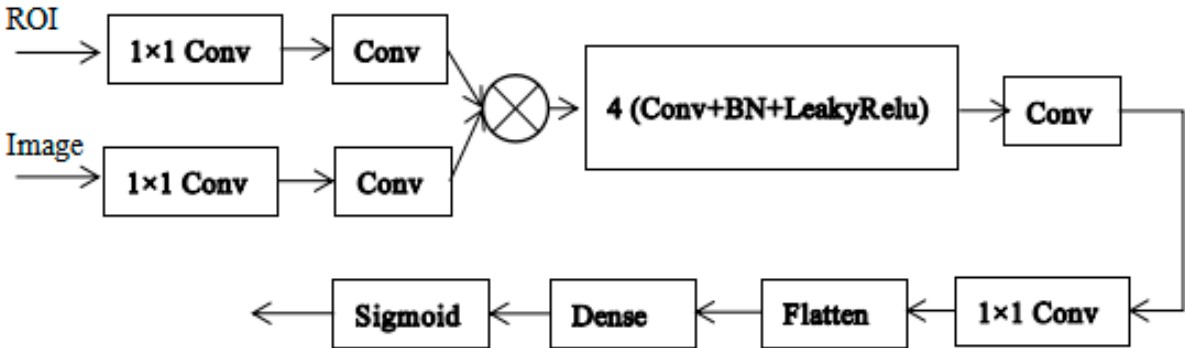

**Figure 5.** The network of the discriminator. The discriminator determines how close the predicted map is to the ground truth by using an evaluation score.

*2.4. Training Strategy of Diff-CoGAN Framework*

In this section, the training strategy of Diff-CoGAN is described in detail, consisting of loss functions and training settings. The aim of the strategy is to guide the proposed Diff-CoGAN framework to optimize its parameters to achieve the best segmentation task.

2.4.1. Loss Functions

Loss functions provide direction for the model's optimization. To describe the loss functions, all the used symbols are first enlisted below:

$X_L^1$ = labeled data used by Generator 1
$X_U^1$ = unlabeled data used by Generator 1
$X_L^2$ = labeled data used by Generator 2
$X_U^2$ = unlabeled data used by Generator 2
$Y_L^1$ = ground truth of the labeled data used by Generator 1
$\overline{Y^1}_L$ = predicted map for the labeled data generated by Generator 1
$\overline{Y^1}_U$ = predicted map for the unlabeled data generated by Generator 1
$Y_L^2$ = ground truth of the labeled data used by Generator 2
$\overline{Y^2}_L$ = predicted map for the labeled data generated by Generator 2
$\overline{Y^2}_U$ = predicted map for the unlabeled data generated by Generator 2

In Diff-CoGAN, the first step is to initialize Generator 1 and Generator 2. This step is implemented under fully supervised learning using the labeled data only. The loss functions for Generator 1 initialization are defined as follows:

$$loss_1 = loss_{bce1} \tag{1}$$

$$loss_{bce1} = -Y^1 \cdot \log\left(\overline{Y^1}\right) - \left(1 - Y^1\right) \cdot \log\left(1 - \overline{Y^1}\right) \tag{2}$$

Similarly, the loss functions for Generator 2 initialization are defined as follows:

$$loss_2 = loss_{bce2} \tag{3}$$

$$loss_{bce2} = -Y^2 \cdot \log\left(\overline{Y^2}\right) - \left(1 - Y^2\right) \cdot \log\left(1 - \overline{Y^2}\right) \tag{4}$$

where *bce* means a binary cross-entropy function.

When the generators' initializations are finished, the discriminator is combined with the generators to construct Diff-CoGAN. The second step is to train Diff-CoGAN using semi-supervised learning. In this step, the loss functions are defined as written below.

For the optimization of the generators, there are 3 losses: segmentation loss, similarity loss, and adversarial loss. Segmentation loss works with the labeled data only. Similarity loss and adversarial loss work with the labeled data and the unlabeled data.

$$Loss_G = loss_{seg} + \alpha loss_{simi} + \beta loss_{adv} \tag{5}$$

$$loss_{seg} = loss_{seg1} + loss_{seg2} \tag{6}$$

$$loss_{seg1} = loss_{dice1} + loss_{bce1} \tag{7}$$

$$loss_{dice1} = 1 - Dice\left(Y^1, \overline{Y^1}\right) = 1 - 2\frac{\left|Y^1 \cap \overline{Y^1}\right|}{|Y^1| + \left|\overline{Y^1}\right|} \tag{8}$$

$$loss_{seg2} = loss_{dice2} + loss_{bce2} \tag{9}$$

$$loss_{dice2} = 1 - Dice\left(Y^2, \overline{Y^2}\right) = 1 - 2\frac{\left|Y^2 \cap \overline{Y^2}\right|}{|Y^2| + \left|\overline{Y^2}\right|} \tag{10}$$

$$loss_{simi} = loss_{dice12L} + loss_{dice12U} \tag{11}$$

$$loss_{dice12L} = 1 - Dice\left(\overline{Y^1}_L, \overline{Y^2}_L\right) \tag{12}$$

$$loss_{dice12U} = 1 - Dice\left(\overline{Y^1}_U, \overline{Y^2}_U\right) \tag{13}$$

$$loss_{adv} = loss_{bce}\left(D\left(X^1_L, \overline{Y^{in}}_L\right), 1\right) + loss_{bce}\left(D\left(X^1_U, \overline{Y^{in}}_U\right), 1\right) \tag{14}$$

$$\overline{Y^{in}}_L = \overline{Y^1}_L \cap \overline{Y^2}_L, \ \overline{Y^{in}}_U = \overline{Y^1}_U \cap \overline{Y^2}_U \tag{15}$$

It should be noted that $\overline{Y^2}_L$ needs to be rotated 180° again (i.e., matrix transposition) when calculating the loss value in functions (10)–(14). Segmentation loss $loss_{seg}$ is used to measure how close the predicted maps generated by the two generators are to the ground truth using fully supervised learning.

Similarity loss, $loss_{simi}$, is used to describe the similarity of the predicted maps from two generators for one data set. According to the loss function definition, two generators can provide mutual segmentation information to each other. Then, the performance of the generators can be improved by mutual guidance between them.

Adversarial loss, $loss_{adv}$, is used to measure the similarity of the predicted maps from the labeled data or unlabeled data to the ground truth. For a generator, it aims to generate a predicted map with high similarity to make the discriminator think that the predicted map is the ground truth. In Diff-CoGAN, when calculating the adversarial loss, the input of the discriminator is the intersection region from the predicted maps generated by the two generators.

Meanwhile, $\alpha$ is the weight of $loss_{simi}$ and $\beta$ is the weight of $loss_{adv}$, and both are set to 0.02 according to the experimental training process in this paper. This is to ensure stable training and better segmentation performance of Diff-CoGAN under different data training settings. For the discriminator's optimization, the loss is the evaluation loss. This loss function is defined as follows:

$$loss_{eva} = loss_{bce}\left(D\left(X^1_L, Y^1_L\right), 1\right) + loss_{bce}\left(D\left(X^1_L, \overline{Y^{in}}_L\right), 0\right) + loss_{bce}\left(D\left(X^1_U, \overline{Y^{in}}_U\right), 0\right) \tag{16}$$

The discriminator should distinguish segmented ROI resources and gives an evaluation score to the input data. For the labeled data, the segmented ROI can be the predicted map from the generators or the ground truth from experts' annotation. For the unlabeled data, the segmented ROI is only from the predicted map. The input data of the discriminator from the ground truth are represented with 1, while 0 represents the input data of the discriminator from the predicted map. It should be noted that the predicted map

involved in the calculation is the intersection region of the predicted maps generated by the two generators.

2.4.2. Training Settings

In this paper, three experiments were conducted: image segmentation based on Diff-CoGAN, segmentation based on the classic GAN, and segmentation based on co-training [7]. There were two steps for segmentation based on Diff-CoGAN:

(1) The first step was to initialize Generator 1 and Generator 2 using the labeled data only. In this step, the optimizer was stochastic gradient descent (SGD), the epoch was set to 50, and batch_size was set to 8. The training aim was to minimize the loss functions, $loss_1$ and $loss_2$.

(2) In the second step, Diff-CoGAN adopted a semi-supervised learning strategy. In the training process, the optimizer for the generators was SGD, and the learning rate was set to 0.01. For the discriminator, the optimizer was Root Mean Squared Propagation (RMSprop), and the learning rate was set to 0.001. The batch_size of the labeled data was set to 8, and the batch_size of the unlabeled data was set to 16. The epoch was set to 100. In the optimization process, the discriminator was trained to minimize $loss_{eva}$, and the generators were trained to minimize $Loss_G$. For the generators' training, the labeled and unlabeled data were firstly used to minimize $loss_{simi}$ and $loss_{adv}$, and $loss_{seg}$ was optimized using the labeled data only. In the training process, the generators were trained ten times and the discriminator was trained once.

There were two steps for segmentation using the classic GAN:

(1) The first step was to initialize Generator 1 and Generator 2 using the labeled data only. This step adopted the initialized generators in Diff-CoGAN.

(2) The second step was to train GAN under semi-supervised learning. There were two generators to realize medical image segmentation in Diff-CoGAN. However, segmentation based on GAN was separated into two experiments: Semi-GAN (G1) which adopted the segmentation approach from [34] and Semi-GAN (G2) from [39]. Semi-GAN (G1) consisted of Generator 1 and the discriminator, and Semi-GAN (G2) consisted of Generator 2 and the discriminator. The optimizer of the generator was SGD, and the learning rate was set to 0.01. The optimizer of the discriminator was RMSprop, and the learning rate was set to 0.001. The batch_size of the labeled data was set to 8, and the batch_size of the unlabeled data was set to 16. The epoch was set to 100.

There were two steps for segmentation using co-training:

(1) The first step was to initialize Generator 1 and Generator 2 using the labeled data only. This step adopted the initialized generators in Diff-CoGAN.

(2) In the second step, the unlabeled data and labeled data were used to train the model together inspired by the work by [7]. In the training process, the optimizer for the generators was SGD, and the learning rate was set to 0.01. The batch_size of the labeled data was set to 8, and the batch_size of the unlabeled data was set to 16. The epoch was set to 100. For these three experiments, the Dropout layer and the Batch normalization layer were imported into the model's structure to prevent overfitting.

## 3. Experiments and Analysis

In this section, the implementation of all experiments is described in detail. Firstly, the datasets and data preprocessing are introduced, then the experimental setup and the design details of the comparative experiments are introduced. Finally, the experimental results are analyzed.

### 3.1. Dataset

In this paper, the data used were the Hippocampus and Spleen images in the Medical Segmentation Decathlon (MSD) dataset. The Hippocampus dataset from the Vanderbilt

University Medical Centre contains three-dimensional (3D) MRI volumes with the corresponding ground truths. The ROI in this paper was the anterior hippocampus. The data source of Spleen was from the Memorial Sloan Kettering Cancer Center. The Spleen dataset contains 61 three-dimensional volumes showing images of CT modality with corresponding ground truths. The ROI of the Spleen dataset was the whole spleen.

Figures 6 and 7 show one Hippocampus data sample and one Spleen data sample, respectively, using a 3D slicer software in three directions: axial, sagittal, and coronal.

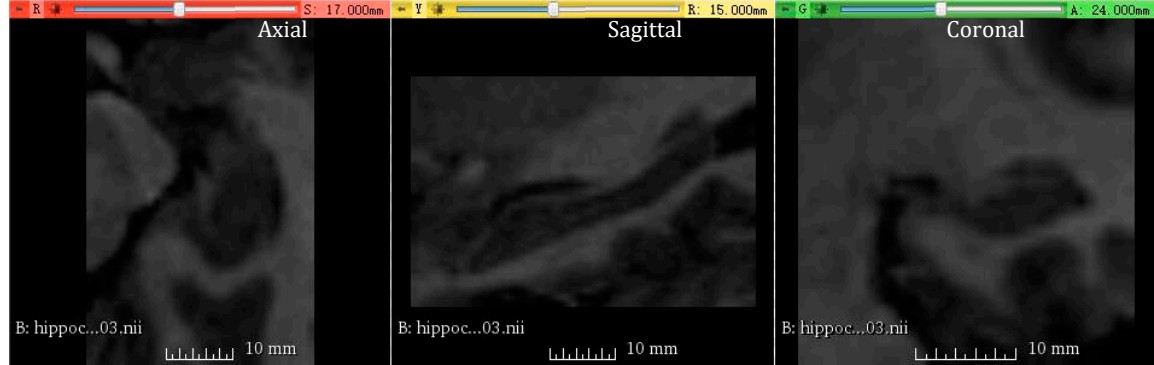

**Figure 6.** A sample of Hippocampus data shown in axial, sagittal, and coronal plane views from left to right. The axial view was chosen in this paper.

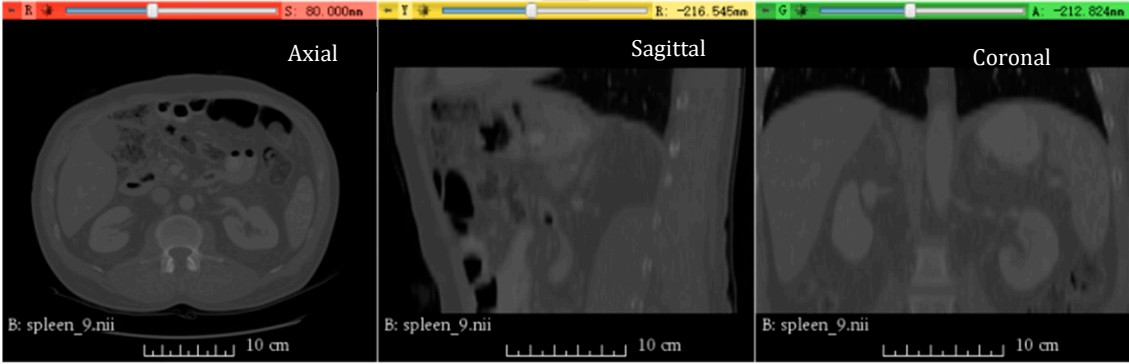

**Figure 7.** A sample of the Spleen data shown in axial, sagittal, and coronal plane views from left to right. The axial view was chosen in this paper.

The 3D MRI volumes and ground truth need to be converted to 2D images because the proposed Diff-CoGAN is a 2D network. The Hippocampus MRI dataset with MRI modality has three directions: axial, sagittal, and coronal. In each direction, the MRI data contain several 2D slices. In this paper, the axial slices were extracted to be used as the 2D images. For the Hippocampus dataset, each data sample has 29~40 slices in the axial direction. During extraction, the slices without a specified ROI were discarded because these slices cannot provide the information of the ROI that we needed. Since the middle slices contain the ROI that we needed, the middle axial slices were used in our experiments. Then, all the 2D slices were separated into the training and testing datasets. The training dataset contained 3000 slices with 2D ground truth, and the testing dataset contains 699 slices with 2D ground truth. The next step was to resize and normalize the gray values to be between 0 and 1. (0,1). The size of the original images of $35 \times 51$ was resized to $64 \times 64$.

After the 3D to 2D data conversion of the Spleen dataset, a total of 1051 2D axial slices with ground truth were gathered. Since the original slices have a large foreground range which may deteriorate the segmentation process, the original slices were cropped. Given that the Spleen was in the upper left part of the axial slice image, the aim of cropping was to acquire the upper left ROI of the original slices. Then, the cropped slices were

resized to $128 \times 128$, and the gray value were normalized to be between 0 and 1. The processed Spleen dataset was further divided into 900 training and 151 testing datasets. After data preprocessing, the training datasets of both Hippocampus and Spleen were further shuffled. Then, the training datasets were separated into the labeled and unlabeled datasets to prepare for semi-supervised learning. The first 100 axial slices were selected as the labeled dataset, while the rest of the axial slices were specified as the unlabeled dataset. Table 2 shows the details of the training dataset settings for the different experiments.

**Table 2.** The training dataset settings. The labeled slices in both Dataset 1 and Dataset 2 were set to 100 slices in all experiments. Meanwhile, the unlabeled slices were varied to investigate how this affected the segmentation performance.

| Experiments | Dataset 1 | Training Dataset Setting (Labeled Slices/Unlabeled Slices) | Dataset 2 | Training Dataset Setting (Labeled Slices/Unlabeled Slices) |
|---|---|---|---|---|
| Semi-supervised learning using co-training Semi-supervised learning using GAN (semi-GAN) Semi-supervised learning using Diff-CoGAN | Hippocampus | 100/100 100/1000 100/2000 100/2900 | Spleen | 100/100 100/400 100/800 |

The labeled data in the training dataset were fixed to 100, while the unlabeled data were increased at varying scales. As recommended by [34], the use of different numbers of unlabeled data was to obtain further insights on the tendency of segmentation performance of the Diff-CoGAN, co-training, and semi-GAN models. There were four training dataset settings for Hippocampus segmentation, and three training dataset settings for Spleen segmentation. Each experiment of Hippocampus segmentation used 100 fixed labeled data points but with different scales of the unlabeled data, including 100, 1000, 2000, and 2900. Meanwhile, the experiments for Spleen segmentation used 100 fixed labeled data points but with different scales of the unlabeled data, including 100, 400, and 800, during training process.

### 3.2. Experimental Setup

The proposed Diff-CoGAN framework was implemented on Keras deep learning API, and all the experiments were conducted on the Kaggle platform. Since the Diff-CoGAN framework has two generators, the segmentation results of co-training, GAN, and Diff-CoGAN were compared according to the generators. Therefore, the segmentation results from Generator 1 in each experiment were compared and named as co-training (G1), semi-GAN (G1), and Diff-CoGAN (G1). Similarly, the segmentation results from Generator 2 in each experiment were compared and named as co-training (G2), semi-GAN (G2), and Diff-CoGAN (G2). As shown in Table 2, there are four training dataset settings for the Hippocampus segmentation task and three training dataset settings for the Spleen segmentation task. The experimental details were set as described in Section 2.3.2.

### 3.3. Performance Evaluation

In this paper, there were four evaluation metrics used: Dice, Intersection of Union (IoU), Hausdorff distance (HD), and average symmetric surface distance (ASD).

Dice: The Dice coefficient is used to evaluate the similarity between two samples. In medical image segmentation, the similarity or overlap between a predicted result and the ground truth can be calculated by the Dice function. The closer the value of Dice to one, the better the segmentation performance, and vice versa. Given two datasets A and B, the Dice index between them is defined as follows [38]:

$$Dice(A, B) = 2 \frac{A \cap B}{|A| + |B|} \tag{17}$$

IoU: IoU is similar to Dice. The range of IoU is from zero to one. An IoU value of one represents completely overlapping samples, and zero represents no overlap. Given two datasets *A* and *B*, the IoU between them is defined as follows [38]:

$$IoU = \frac{|A \cap B|}{|A \cup B|} \tag{18}$$

Hausdorff distance: It describes the similarity between two sets of points, that is, the distance between the boundaries of the ground truth and a predicted result. It is sensitive to the segmentation of the boundary. Hausdorff distance is defined as follows [38]:

$$H = \max \left( \max_{i \in pr} \left( \min_{j \in gt} (d(i,j)) \right), \max_{j \in gt} \left( \min_{i \in pr} (d(i,j)) \right) \right) \tag{19}$$

where *i* and *j* are the points belonging to different sets. Additionally, *d* represents the distance between *i* and *j*.

ASD: ASD describes the similarity between two sets of points. It is similar to Hausdorff distance, but ASD is the average distance between two sets of points [40]:

$$ASD = \frac{1}{|S(A)| + |S(B)|} \left( \sum_{a \in S(A)} \min_{b \in S(B)} \|a - b\| + \sum_{b \in S(B)} \min_{a \in S(A)} \|a - b\| \right) \tag{20}$$

where $S(A)$ and $S(B)$ represent the points of boundary for set A and set B, and a and b are the points belonging to $S(A)$ and $S(B)$.

*3.4. Results*

The results are presented in two subsections, one for each type of dataset, that is, the Hippocampus dataset and the Spleen dataset.

3.4.1. Medical Image Segmentation Using Hippocampus Dataset

All the experimental results using Dice, IoU, HD, and ASD are presented in Tables 3–7. In co-training, classic GAN, and Diff-CoGAN, the first step was to initialize Generator 1 and Generator 2 using the labeled data only. Table 3 presents the Dice, IoU, HD, and ASD values for the Hippocampus dataset when using 100 fixed labeled data points to train Generator 1 and Generator 2, respectively. The experiment of Generator 1 initialization was named *seg1only*, and the experiment of Generator 2 initialization was named *seg2only*.

**Table 3.** The Dice, IoU, HD, and ASD values show that Generator 2 performs better than Generator 1. The experiment was conducted using only 100 labeled data points of the Hippocampus dataset.

| Initialization | Dice | IoU | HD | ASD |
|---|---|---|---|---|
| *seg1only* | 0.460 (0.045) | 0.072 (0.003) | 28.209 (111.717) | 10.639 (21.102) |
| *seg2only* | 0.731 (0.018) | 0.285 (0.016) | 10.079 (21.403) | 3.459 (1.715) |

From Table 3, both Dice and IoU values of *seg1only* are lower than *seg2only*, indicating that Generator 2 has better segmentation accuracy. Furthermore, the HD value and ASD value of *seg1only* are higher than *seg2only*, also showing that Generator 2 reduces the boundary error during segmentation better than Generator 1. According to the network design of Generator 1 and Generator 2, the difference is the encoder structure in which Generator 1 adopts DenseNet without pre-training, while Generator 2 adopts trained VGG16 with transfer learning. Generator 2 has a simpler structure but achieves better performance partly due to the transfer learning strategies.

**Table 4.** The Dice and IoU values of co-training, semi-GAN, and Diff-CoGAN using the Hippocampus dataset. Unlike semi-GAN, co-training and Diff-CoGAN show an improvement in segmentation as more unlabeled data are added to the training.

| Data Setting | Dice | | | | | | IOU | | | | | |
| --- | --- | --- | --- | --- | --- | --- | --- | --- | --- | --- | --- | --- |
| | Co-Training | | Semi-GAN | | Diff-CoGAN | | Co-Training | | Semi-GAN | | Diff-CoGAN | |
| | G1 | G2 | G1 | G2 | G1 | G2 | G1 | G2 | G1 | G2 | G1 | G2 |
| 100/100 | 0.774 (0.015) | 0.785 (0.017) | 0.774 (0.015) | 0.783 (0.018) | 0.780 (0.014) | 0.805 (0.012) | 0.543 (0.021) | 0.551 (0.022) | 0.543 (0.022) | 0.547 (0.023) | 0.545 (0.021) | 0.578 (0.019) |
| 100/1000 | 0.793 (0.014) | 0.796 (0.019) | 0.783 (0.015) | 0.802 (0.012) | 0.797 (0.013) | 0.808 (0.011) | 0.576 (0.020) | 0.582 (0.024) | 0.534 (0.022) | 0.581 (0.019) | 0.586 (0.019) | 0.589 (0.018) |
| 100/2000 | 0.801 (0.01) | 0.811 (0.011) | 0.686 (0.028) | 0.802 (0.013) | 0.804 (0.012) | 0.812 (0.011) | 0.593 (0.018) | 0.600 (0.017) | 0.424 (0.027) | 0.581 (0.019) | 0.593 (0.018) | 0.601 (0.017) |
| 100/2900 | **0.804** (0.011) | **0.814** (0.011) | **0.790** (0.014) | **0.806** (0.011) | **0.805** (0.012) | **0.814** (0.011) | **0.595** (0.018) | **0.606** (0.017) | **0.576** (0.020) | **0.590** (0.018) | **0.597** (0.018) | **0.605** (0.017) |

**Table 5.** The HD and ASD values using the Hippocampus dataset show that Diff-CoGAN performs better than co-training and semi-GAN when processing boundaries during segmentation.

| Data Setting | HD | | | | | | ASD | | | | | |
| --- | --- | --- | --- | --- | --- | --- | --- | --- | --- | --- | --- | --- |
| | Co-Training | | Semi-GAN | | Diff-CoGAN | | Co-Training | | Semi-GAN | | Diff-CoGAN | |
| | G1 | G2 | G1 | G2 | G1 | G2 | G1 | G2 | G1 | G2 | G1 | G2 |
| 100/100 | 8.276 (12.405) | 7.384 (8.190) | 7.921 (9.190) | 7.265 (8.200) | 8.081 (14.777) | 7.264 (7.756) | 3.148 (1.425) | 2.724 (1.154) | 3.087 (1.269) | 2.606 (1.081) | 3.037 (1.628) | 2.624 (0.906) |
| 00/1000 | 6.813 (9.286) | 6.550 (7.673) | 7.316 (13.288) | 6.807 (7.847) | 6.537 (7.426) | 6.533 (6.917) | 2.596 (1.626) | 2.468 (1.398) | 2.473 (1.490) | 2.571 (1.017) | 2.420 (1.076) | 2.458 (1.032) |
| 100/2000 | 6.117 (8.939) | 5.990 (7.354) | 8.852 (20.004) | 7.034 (8.010) | 5.966 (7.420) | 5.963 (7.182) | 2.054 (0.626) | 2.050 (0.837) | 2.902 (1.936) | 2.728 (1.122) | 2.044 (0.549) | 2.043 (0.885) |
| 100/2900 | 5.902 (7.203) | 5.777 (6.683) | 6.163 (8.888) | 6.038 (7.518) | 5.832 (6.506) | 5.765 (6.656) | 1.952 (0.815) | 1.866 (0.340) | 2.110 (1.002) | 2.030 (0.841) | 1.917 (0.506) | 1.863 (0.420) |

**Table 6.** The HD and ASD values of Diff-CoGAN with and without intersection prior to the input of the discriminator. Using the Hippocampus dataset, the intersection operation shows that it is able to reduce the boundary errors in segmentation. Diff-CoGAN with intersection consistently shows lower values of ASD and HD compared to Diff-CoGAN without intersection.

| Data Setting | HD | | | | ASD | | | |
| --- | --- | --- | --- | --- | --- | --- | --- | --- |
| | Diff-CoGAN (without) | | Diff-CoGAN (with) | | Diff-CoGAN (without) | | Diff-CoGAN (with) | |
| | G1 | G2 | G1 | G2 | G1 | G2 | G1 | G2 |
| 100/100 | 8.433 (12.695) | 7.306 (8.646) | 8.081 (14.777) | 7.264 (7.756) | 3.300 (1.583) | 2.660 (1.136) | 3.037 (1.628) | 2.624 (0.906) |
| 100/1000 | 7.251 (8.028) | 7.108 (6.887) | 6.537 (7.426) | 6.533 (6.917) | 2.894 (1.255) | 2.849 (0.962) | 2.420 (1.076) | 2.458 (1.032) |
| 100/2000 | 6.678 (10.805) | 6.261 (6.084) | 5.966 (7.420) | 5.963 (7.182) | 2.233 (0.845) | 2.282 (0.533) | 2.044 (0.549) | 2.043 (0.885) |
| 100/2200 | 8.256 (15.609) | 5.932 (6.790) | 6.045 (8.258) | 5.880 (6.366) | 2.420 (1.184) | 2.009 (0.485) | 2.031 (0.554) | 1.967 (0.413) |
| 100/2400 | 5.928 (7.087) | 5.851 (6.326) | 5.905 (6.381) | 5.851 (6.993) | 1.970 (0.646) | 1.945 (0.423) | 1.979 (0.492) | 1.929 (0.471) |
| 100/2900 | **5.833** (6.151) | **5.795** (6.918) | **5.832** (6.506) | **5.765** (**6.656**) | **1.942** (0.435) | **1.877** (0.511) | **1.917** (0.506) | **1.863** (**0.420**) |

**Table 7.** The metric values for the Spleen dataset using only 100 labeled data points for training. The use of VGG16 in Generator 2 (*seg2only*) shows that it performs better than Generator 1 (*seg1only*) with a higher segmentation accuracy and is able to reduce the boundary error during segmentation.

| Initialization | Dice | IoU | HD | ASD |
|:---:|:---:|:---:|:---:|:---:|
| *seg1only* | 0.161 (0.041) | 0.019 (0.001) | 83.886 (136.112) | 40.423 (68.744) |
| *seg2only* | **0.886** (0.007) | **0.410** (0.018) | **28.307** (188.846) | **7.665** (10.764) |

In Tables 4 and 5, the results of co-training, semi-GAN, and Diff-CoGAN are presented. For a clearer comparison and analysis, G1 presents the results generated by Generator 1, and G2 presents the results generated by Generator 2 in each experiment. Co-training achieved segmentation using mutual segmentation information between the two generators. Semi-GAN achieved segmentation by using adversarial training in classic GAN. Diff-CoGAN achieved segmentation by using mutual information guidance and adversarial training between the generators and discriminator.

For Hippocampus segmentation, Table 4 shows the Dice values and IoU values using four data settings, and the value shown in parenthesis are the distribution variations of the Dice values and IoU values. The highest Dice values are all produced using data setting 100/2900 for co-training (G1 with 0.804 and G2 with 0.814), Semi-GAN (G1 with 0.790 and G2 with 0.806), and Diff-CoGAN (G1 with 0.805 and G2 with 0.814). It means the unlabeled data can improve the segmentation performance in semi-supervised learning. When compared at the data setting level, Diff-CoGAN (G1) improved the Dice value by 0.6%, 0.4%, 0.3%, and 0.1% compared to co-training (G1) for the data settings 100/100, 100/1000, 100/2000, and 100/2900, respectively. At the same time, Diff-CoGAN (G1) improved the Dice value by 0.6%, 1.4%, 11.8%, and 1.5% compared to semi-GAN (G1) for the data settings 100/100, 100/1000, 100/2000, and 100/2900, respectively. Similarly, Diff-CoGAN (G2) improved the Dice value by 2%, 1.2%, 0.1%, and 0% compared to co-training (G2) for the data settings 100/100, 100/1000, 100/2000, and 100/2900, respectively. Additionally, Diff-CoGAN (G2) improved the Dice value by 2.2%, 0.6%, 10%, and 0.8% compared to Semi-GAN (G2) for the data settings 100/100, 100/1000, 100/2000, and 100/2900, respectively. By comparing Diff-CoGAN with semi-GAN, it shows that Diff-CoGAN could improve the segmentation performance by adopting mutual guidance and adversarial training. By comparing Diff-CoGAN with co-training, it shows that the adversarial training of Diff-CoGAN could improve segmentation performance significantly when using less unlabeled data.

It is also worth noting that the Dice values of Diff-CoGAN and co-training improved as more unlabeled data was added to the training dataset. However, for semi-GAN (G1), the Dice value dropped to the lowest value of 0.686 at the data setting 100/2000, while the Dice value of semi-GAN (G2) was retained at 0.802 at the data settings 100/1000 and 100/2000. The co-training and Diff-CoGAN frameworks showed a more stable performance, and more unlabeled data also helped increase their image segmentation.

Table 4 shows that the highest IoU values are all produced using the data setting 100/2900 for co-training (G1 with 0.595 and G2 with 0.606), semi-GAN (G1 with 0.576 and G2 with 0.590), and Diff-CoGAN (G1 with 0.597 and G2 with 0.605). When compared at the data setting level, Diff-CoGAN (G1) improved the IoU values by 0.2%, 1%, 0%, and 0.2% compared to co-training (G1) for the data settings 100/100, 100/1000, 100/2000, and 100/2900, respectively. Diff-CoGAN (G1) improved the IoU values by 0.2%, 5.2%, 16.9%, and 2.1% compared to semi-GAN1 for the data settings 100/100, 100/1000, 100/2000, and 100/2900, respectively. Similarly, Diff-CoGAN (G2) improved the IoU value by 2.7%, 0.7%, 0.1%, and −0.1% compared to co-training (G2) for the data settings 100/100, 100/1000, 100/2000, and 100/2900, respectively. Additionally, Diff-CoGAN (G2) improved the IoU value by 3.1%, 0.8%, 2%, and 1.5% compared to semi-GAN (G2) for the data settings 100/100, 100/1000, 100/2000, and 100/2900, respectively. The IoU values have the same

regularity as the Dice values. The IoU values also showed that segmentation performed by Diff-CoGAN was improved consistently as more unlabeled data were added. On the other hand, the values of IoU of semi-GAN reduced or remained the same when more unlabeled data were added. For example, the IoU value of semi-GAN (G1) reduced to 0.424 and the value of semi-GAN (G2) was retained at 0.581 at the data setting 100/2000. Additionally, Diff-CoGAN performed better when less unlabeled data were added to the training process compared to co-training.

Besides, the Dice value for *seg1only* is 0.460 in Table 3, which is lower than the Dice values of G1 for all data settings in the different experiments in Table 4. The Dice value for *seg2only* is 0.731 in Table 3, which is lower than the Dice values of G2 for all data settings in the different experiments in Table 4. Similarly, the IoU value for *seg1only* is 0.072 in Table 3, which is lower than the IoU values of G1 for all data settings in Table 4. The IoU value for *seg2only* is 0.285 in Table 3, which is lower than the IoU values of G2 for all data settings in Table 4. These results indicate that the mutual segmentation information between the generators in the co-training experiment can improve the segmentation performance compared to *seg1only* or *seg2only*. Similarly, the added discriminator in semi-GAN can improve the segmentation performance, which also means adversarial training between the generator and the discriminator can play a part in the model training process compared to *seg1only* or *seg2only*.

Table 5 displays the HD and ASD metrics' evaluation for the four data settings for segmentation using the Hippocampus dataset. The values shown in parenthesis are the distribution variations of the HD and ASD values. The best HD values were produced for the data setting 100/2900 in which Diff-CoGAN (G1) achieved 5.832, a reduction of 7% compared to co-training (G1) and 33.1% compared to semi-GAN (G1), and Diff-CoGAN (G2) achieved 5.765, a reduction of 1.2% compared to co-training (G2) and 27.3% compared to semi-GAN (G2). Unlike the semi-GAN models, the values of HD for co-training and Diff-CoGAN continuously reduced as more unlabeled data were added. Furthermore, the variation in Diff-CoGAN scored the lowest value compared to co-training and semi-GAN, which means that the HD values' distribution in Diff-CoGAN (G1) has a lower degree of dispersion (6.506) than co-training (G1) (7.203) and semi-GAN (G1) (8.888). Diff-CoGAN (G2) also has fewer dispersion (6.656) than co-training (G2) (6.683) and semi-GAN (G2) (7.518). According to the table, Diff-CoGAN has stable performance as shown in its HD values, indicating that the proposed Diff-CoGAN has better effectiveness in boundary processing.

The segmentation performance of the proposed Diff-CoGAN is also supported by the ASD values presented in Table 5. The regularities of the ASD values' distribution are similar to the HD values. The best ASD values were generated in the data setting 100/2900 in which Diff-CoGAN (G1) achieved 1.917, a reduction of 3.5% compared to co-training (G1) and 19.3% compared to semi-GAN (G1). Additionally, Diff-CoGAN (G2) achieved an ASD value of 1.863, which is 16.7% smaller compared to semi-GAN (G2) and 0.3% smaller compared to co-training (G2). Thus, the results consistently show that Diff-CoGAN performs better than co-training and classical GAN.

The HD values for *seg1only* and *seg2only* are 28.209 and 10.079, respectively, in Table 3. Both are higher compared to the HD values for co-training, semi-GAN, and Diff-CoGAN in Table 5 for all data settings. Similarly, the ASD values for *seg1only* and *seg2only* are 10.639 and 3.459, respectively, in Table 3, which are higher than the ASD values of the different experiments for all data settings in Table 5. These results indicate that the mutual segmentation guidance between the generators (co-training) and added discriminator (semi-GAN) could reduce the boundary error in segmentation.

In Diff-CoGAN, the input for the discriminator was the segmented ROI, which was obtained from the intersection of the predicted maps generated by the two generators and the ground truth annotated by experts. The purpose of intersection was to reduce the boundary errors for the predicted maps in the training process. Table 6 shows the results of Diff-CoGAN with and without intersection in the training process, namely

Diff-CoGAN(with) and Diff-CoGAN(without). In Table 6, the HD values are lower for G1(with) than G1(without) in the data settings from 100/100 to 100/2200. It means that the intersection operation could reduce the boundary errors in segmentation. Similarly, the HD values for G2(with) are also lower than G2(without) in the data settings from 100/100 to 100/2200. The ASD values have the same distribution as the HD values. However, when using the data settings 100/2400 and 100/2900, the HD and ASD values of Diff-CoGAN(with) are very close to Diff-CoGAN(without). Thus, the effect of intersection is obvious when using unlabeled data less than 2200. Table 6 shows the effect of intersection, which can improve the segmentation performance.

3.4.2. Medical Image Segmentation Using Spleen Dataset

For the Spleen segmentation, Table 7 shows the Dice, IoU, HD, and ASD values for *seg1only* and *seg2only* as the base comparison to illustrate the effects of unlabeled data, interaction effect between generators, and adversarial training in the model training process. The initialization results presented in Table 7 support the findings of Table 3. When using the Spleen dataset, Generator 2 (*seg2only*) shows a higher accuracy at segmentation with higher values of the Dice and IoU metrics compared to Generator 1 (*seg1only*). The distance metrics of HD and ASD for Generator 2 (*seg2only*) are also lower than Generator 1 (*seg1only*), signifying that the use of VGG16 in Generator 2 is better at reducing the boundary error during segmentation. This is partly because VGG16 was already trained with the large dataset ImageNet, and the trained VGG16 has a higher feature extraction capacity that can be used and fine-tuned with transfer learning for medical image segmentation.

Table 8 shows the Dice and IoU values using three data settings, and the values shown in parenthesis are the variations in the Dice and IoU distributions. Overall, the Dice values improved slightly as the unlabeled data increased. The highest Dice values were all produced using the data setting 100/900 for co-training (G1) (0.933), co-training (G2) (0.947), semi-GAN (G1) (0.899), semi-GAN (G2) (0.947), Diff-CoGAN (G1) (0.914), and Diff-CoGAN (G2) (0.948). When compared at the data setting level, Diff-CoGAN (G1) improved the Dice value by 2.6%, 1.0%, and −0.3% compared to co-training (G1) for the data settings 100/100, 100/400, and 100/900, respectively. At the same time, Diff-CoGAN (G1) improved the Dice value by 0.7%, 1.8%, and 3.1% compared to semi-GAN (G1) for the data settings 100/100, 100/400, and 100/900, respectively. These results mean that Diff-CoGAN achieves slightly better segmentation performance by using mutual guidance and adversarial training when using less unlabeled data.

**Table 8.** The Dice and IoU values for the Spleen dataset. As more unlabeled data are added, the IoU values for all three models increase. Diff-CoGAN outperforms both co-training and semi-GAN due to the mutual guidance between the generators and the adversarial training of the two generators and the discriminator.

| Data Setting | Dice | | | | | | IoU | | | | | |
| --- | --- | --- | --- | --- | --- | --- | --- | --- | --- | --- | --- | --- |
| | Co-Training | | Semi-GAN | | Diff-CoGAN | | Co-Training | | Semi-GAN | | Diff-CoGAN | |
| | G1 | G2 | G1 | G2 | G1 | G2 | G1 | G2 | G1 | G2 | G1 | G2 |
| 100/100 | 0.874 (0.019) | 0.943 (0.002) | 0.893 (0.010) | 0.944 (0.002) | 0.900 (0.014) | 0.944 (0.002) | 0.714 (0.029) | 0.812 (0.008) | 0.731 (0.021) | 0.813 (0.008) | 0.747 (0.023) | 0.815 (0.007) |
| 100/400 | 0.904 (0.010) | 0.945 (0.002) | 0.896 (0.011) | 0.947 (0.002) | 0.914 (0.010) | 0.946 (0.002) | 0.760 (0.016) | 0.820 (0.006) | 0.744 (0.020) | 0.822 (0.007) | 0.771 (0.017) | 0.828 (0.006) |
| 100/900 | **0.933** (0.003) | **0.947** (0.002) | **0.899** (0.013) | **0.947** (0.002) | **0.930** (0.003) | **0.948** (0.002) | **0.792** (0.013) | **0.829** (0.006) | **0.757** (0.020) | **0.828** (0.006) | **0.800** (0.016) | **0.832** (0.005) |

The IoU values in Table 8 are also improved as more unlabeled data were added. The highest values were all produced using the data setting 100/900 for co-training (G1) (0.792), co-training (G2) (0.829), semi-GAN (G1) (0.757), semi-GAN (G2) (0.828), Diff-CoGAN (G1) (0.800), and Diff-CoGAN (G2) (0.832). When compared at the data setting level, Diff-CoGAN (G1) improved the IoU value by 1.6%, 2.7%, and 4.3% compared to semi-GAN

(G1) for the data settings 100/100, 100/400, and 100/900, respectively. Diff-CoGAN (G1) improved the IoU value by 3.3%, 1.1%, and 0.8% compared to co-training (G1) for the data settings 100/100, 100/400, and 100/900, respectively. At the same time, Diff-CoGAN (G2) also improved the IoU value to some extent compared to co-training (G2) and semi-GAN (G2). By analyzing the results of Dice and IoU, Diff-CoGAN was shown to perform better than co-training, which means the adversarial training between the generators and discriminator could improve the segmentation performance. Furthermore, Diff-CoGAN could improve performance by adopting mutual guidance when compared to semi-GAN.

The Dice value Is 0.161 for *seg1only* in Table 7, which is lower than the Dice values of G1 for all data settings in Table 8. The Dice value for *seg2only* is 0.886 in Table 7, which is lower than the Dice values of G2 for all data settings in Table 8. Similarly, the IoU value for *seg1only* is 0.019 in Table 7, which is lower than the IoU values of G1 for all data settings in Table 8. The IoU value for *seg2only* is 0.410 in Table 7, which is lower than the IoU values of G2 for all data settings in Table 8. These comparisons show that the mutual segmentation information of the generators and the adversarial training between the generators and the discriminator are active in the training.

Table 9 shows the HD and ASD values using three data settings, and the values shown in parenthesis are the variations in the HD and ASD values' distribution. As the number of unlabeled data increases, the HD values decrease. The lowest HD values were produced in the data setting 100/900 in which Diff-CoGAN (G1) scored 7.253, showing a reduction of 7.898 when compared to semi-GAN (G1) at 15.151 and a reduction of 0.091 when compared to co-training (G1) at 7.344. Meanwhile, Diff-CoGAN (G2) achieved 6.369, which was lower by 3.206 when compared to semi-GAN (G2) at 9.575 and lower by 1.001 when compared to co-training (G2). The results indicate that Diff-CoGAN produces more accurate boundary segmentation compared to semi-GAN and co-training.

**Table 9.** The HD and ASD values for Spleen. The lowest HD value achieved by Diff-CoGAN shows that it produces more accurate boundary segmentation compared to semi-GAN and co-training.

| Data Setting | HD | | | | | | ASD | | | | | |
|---|---|---|---|---|---|---|---|---|---|---|---|---|
| | Co-Training | | Semi-GAN | | Diff-CoGAN | | Co-Training | | Semi-GAN | | Diff-CoGAN | |
| | G1 | G2 | G1 | G2 | G1 | G2 | G1 | G2 | G1 | G2 | G1 | G2 |
| 100/ 100 | 21.244 (221.23) | 15.078 (127.9) | 18.502 (185.37) | 14.080 (111.72) | 17.587 (155.476) | 13.351 (94.51) | 6.226 (21.12) | 3.851 (5.170) | 5.605 (17.15) | 3.602 (3.919) | 5.079 (15.048) | 3.400 (2.285) |
| 100/ 400 | 12.336 (140.521) | 9.053 (31.39) | 15.661 (154.10) | 12.187 (64.307) | 12.099 (152.556) | 8.740 (31.70) | 3.633 (10.94) | 2.866 (1.554) | 4.551 (9.385) | 3.660 (3.401) | 3.570 (11.839) | 2.729 (1.189) |
| 100/ 900 | **7.344** (86.641) | **7.370** (76.36) | **15.151** (268.41) | **9.575** (102.35) | **7.253** (58.916) | **6.369** (32.36) | **2.300** (6.600) | **2.043** (3.590) | **3.040** (9.370) | **2.485** (5.063) | **2.307** (3.177) | **1.849** (0.630) |

The HD values are 83.886 and 28.307 for *seg1only* and *seg2only*, respectively, as seen in Table 7. These values are highest when compared to the corresponding metrics for all experiments in Table 9. This result indicates that the mutual guidance between the generators and the adversarial training between the generator and the discriminator have an effect on reducing boundary errors.

It can also be seen from Table 9 that the best ASD values are generated in the data setting 100/900. Diff-CoGAN (G1) achieved 2.307, a reduction of 73.3% when compared to semi-GAN (G1), and Diff-CoGAN (G2) achieved 1.849, a reduction of 63.6% when compared to semi-GAN (G2) and 19.4% when compared to co-training (G2). Again, Diff-CoGAN shows a better performance at segmenting the ROI boundary compared to semi-GAN and co-training.

Table 10 shows the results of Diff-CoGAN with and without intersection in the training process. In Table 10, the HD values are lower for Co-GAN1(with) than Co-GAN1(without) in each data setting, except the data setting 100/100. It means the intersection operation could reduce the boundary error in segmentation when unlabeled data are increased. Similarly, the HD values for Co-GAN2(with) are also lower than Co-GAN2(without) in

all data settings. In Table 10, the ASD values for Co-GAN1(with) are lower than Co-GAN1(without) in all data settings. The same is true for Co-GAN2. Table 10 further demonstrates that the intersection operation can improve the segmentation performance.

**Table 10.** The HD and ASD of Diff-CoGAN with and without intersection for the Spleen dataset. Similar to the experimental results of the Hippocampus dataset, the HD and ASD values demonstrate that the intersection operation reduces the boundary error and improves segmentation.

| Data Setting | HD | | | | ASD | | | |
| --- | --- | --- | --- | --- | --- | --- | --- | --- |
| | Diff-CoGAN (without) | | Diff-CoGAN (with) | | Diff-CoGAN (without) | | Diff-CoGAN (with) | |
| | G1 | G2 | G1 | G2 | G1 | G2 | G1 | G2 |
| 100–100 | 15.065 (94.445) | 16.065 (127.375) | 17.587 (155.476) | 13.351 (94.517) | 5.185 (13.472) | 4.035 (5.021) | 5.079 (15.048) | 3.400 (2.285) |
| 100–400 | 16.238 (218.456) | 9.224 (21.845) | 12.099 (152.556) | 8.740 (31.700) | 4.408 (17.928) | 2.963 (1.128) | 3.570 (11.839) | 2.729 (1.189) |
| 100–900 | 11.667 (180.558) | 7.839 (91.117) | 7.253 (58.916) | 6.369 (32.365) | 2.697 (6.920) | 2.162 (5.497) | 2.307 (3.177) | 1.849 (0.630) |

Examples of the segmentation results using 100 labeled data points and 2900 unlabeled data points from the Hippocampus dataset are shown in Figure 8. The semi-supervised models and their corresponding Dice values are presented in each segmentation result. The segmented regions (in white) overlap the ground truth (in gray). The figure shows that the results of Diff-CoGAN are closer to the ground truth compared to semi-GAN.

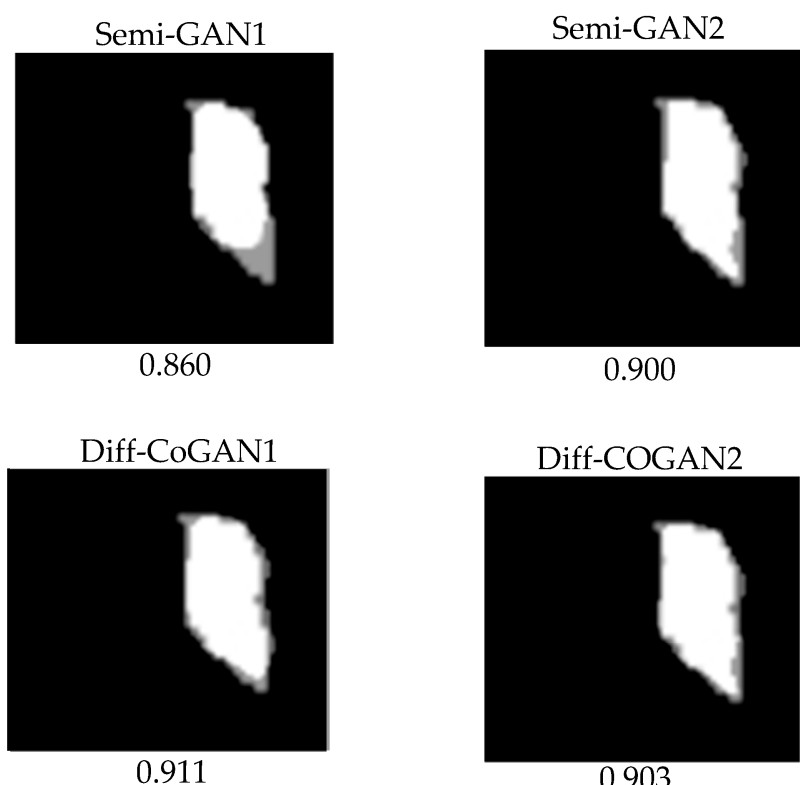

**Figure 8.** Examples of the segmentation results for the Hippocampus dataset. The white regions are the segmented results, and the gray portions are the ground truth.

## 4. Discussion

The results demonstrated that our proposed semi-supervised learning framework, Diff-CoGAN, can leverage unlabeled data to achieve better segmentation accuracy than recent semi-supervised GAN [34], modified GAN [39] and semi-supervised co-training [7].

As more unlabeled data were added to the training, the Dice and IoU values of Diff-CoGAN steadily increased, indicating better accuracy, and the ASD and HD values decreased, signifying less boundary errors. The experimental evidence is shown in Tables 4 and 5 for the Hippocampus dataset. Meanwhile, Tables 8 and 9 showcase the evidence for the Spleen dataset. The adversarial training between the generators and the discriminator was shown to improve segmentation accuracy, as can be seen from Table 4, when Diff-CoGAN was compared with semi-supervised co-training. We further conducted paired *t*-tests on the four metrics (see Table 11) to determine whether the difference between the semi-supervised models is statistically significant. Based on Table 11, Diff-CoGAN achieves higher segmentation accuracy with statistically significant difference ($p < 0.05$) when compared to co-training, except for the data setting 100–2900, while the IoU values show that Diff-CoGAN has better performance with statistical difference ($p < 0.05$) when compared to co-training for all data settings. The adversarial training between the generators and the discriminator in Diff-CoGAN is able to generate predicted maps that are closer to the ground truth compared to semi-supervised co-training. When the results of Diff-CoGAN were compared to semi-GAN, the two generators used in Diff-CoGAN improved the segmentation accuracy (see Table 4) and reduced the boundary segmentation errors (see Table 5). The paired *t*-test ($p < 0.05$) for all four metrics also confirmed that Diff-CoGAN achieved higher segmentation accuracy and improved boundary segmentation with significant difference, particularly when the unlabeled data were increased. This suggests that the strategy of mutual guidance between the two generators can effectively provide segmentation information from the unlabeled data to generate predicted maps that are closer to the ground truth. Additionally, the two generators are optimized under one discriminator' guidance during the training, which further facilitates the generation of better predicted maps.

**Table 11.** Paired *t*-test between Diff-CoGAN, co-training, and semi-GAN using the Hippocampus dataset. In general, the *p*-values for the four metrics are less than 0.5; thus, we can accept the alternative hypothesis that there are statistically significant differences.

| Data Setting | Dice | | | | IoU | | | | HD | | | | ASD | | | |
|---|---|---|---|---|---|---|---|---|---|---|---|---|---|---|---|---|
| | Co-Training | | SemiGAN | | Co-Training | | SemiGAN | | Co-Training | | SemiGAN | | Co-Training | | SemiGAN | |
| | G1 | G2 | G1 | G2 | G1 | G2 | G1 | G2 | G1 | G2 | G1 | G2 | G1 | G2 | G1 | G2 |
| 100–100 | <0.05 | <0.01 | <0.05 | <0.01 | >0.05 | <0.01 | >0.05 | <0.01 | >0.05 | >0.05 | >0.05 | >0.05 | <0.05 | <0.05 | >0.05 | >0.05 |
| 100–1000 | <0.05 | <0.01 | <0.01 | <0.01 | <0.01 | <0.01 | <0.01 | <0.01 | <0.01 | >0.05 | <0.01 | <0.01 | <0.01 | >0.05 | >0.05 | <0.01 |
| 100–2000 | <0.05 | <0.05 | <0.01 | <0.01 | >0.05 | <0.01 | >0.05 | <0.01 | <0.05 | >0.05 | <0.01 | <0.01 | >0.05 | >0.05 | <0.01 | <0.01 |
| 100–2900 | >0.05 | >0.05 | <0.01 | <0.01 | <0.01 | <0.01 | <0.01 | <0.01 | >0.05 | >0.05 | <0.01 | <0.01 | >0.05 | >0.05 | <0.01 | <0.01 |

For the segmentation of the Spleen dataset, the Dice and IoU values (See Table 8) were close between Diff-CoGAN and the other semi-supervised models. A paired *t*-test was also conducted, and the results are presented in Table 12. In terms of segmentation accuracy, Diff-CoGAN showed better performance compared to co-training using Generator 2 ($p < 0.05$ for *t*-test). However, there were no significant differences when Diff-CoGAN was compared to co-training using Generator 2 and semi-GAN models. Table 8 demonstrates that Diff-CoGAN manages to reduce the boundary segmentation errors as more unlabeled data are added. This finding is also supported by the paired *t*-test ($p < 0.05$), as shown by the HD and ASD metrics in Table 12.

**Table 12.** Paired *t*-test between Diff-CoGAN, co-training, and semi-GAN using the Spleen dataset.

| Data Setting | Dice | | | | IoU | | | | HD | | | | ASD | | | |
|---|---|---|---|---|---|---|---|---|---|---|---|---|---|---|---|---|
| | Co-Training | | SemiGAN | | Co-Training | | SemiGAN | | Co-Training | | SemiGAN | | Co-Training | | SemiGAN | |
| | G1 | G2 | G1 | G2 | G1 | G2 | G1 | G2 | G1 | G2 | G1 | G2 | G1 | G2 | G1 | G2 |
| 100–100 | <0.01 | >0.05 | >0.05 | >0.05 | <0.01 | >0.05 | <0.05 | >0.05 | <0.05 | <0.05 | >0.05 | >0.05 | <0.01 | >0.05 | <0.05 | >0.05 |
| 100–400 | <0.05 | >0.05 | <0.01 | >0.05 | <0.01 | <0.01 | <0.01 | <0.01 | >0.05 | <0.05 | <0.01 | <0.01 | <0.05 | <0.05 | <0.01 | <0.01 |
| 100–900 | <0.05 | >0.05 | >0.05 | <0.05 | <0.01 | >0.05 | >0.05 | <0.05 | <0.05 | <0.05 | <0.01 | <0.01 | >0.05 | <0.01 | <0.01 | <0.05 |

The intersection of the predicted maps produced by the generators in our proposed Diff-CoGAN further improves the accuracy of segmented boundaries (See Tables 6 and 10). Table 13 shows the paired *t*-test for Diff-CoGAN with and without intersection operation for both Hippocampus and Spleen datasets. The values ($p < 0.05$) for both Hippocampus and Spleen datasets show that there is a significant difference for all four metrics. While the difference is more significant when less than 2000 unlabeled data points were used for the Hippocampus dataset, the Spleen dataset shows more significant difference as more unlabeled data were used in the training. This is because the intersection of the predicted maps has high confidence for the target region, thereby avoiding segmentation errors during the iterative training process.

**Table 13.** Paired *t*-test between Diff-CoGAN with intersection and Diff-CoGAN without intersection operation. The *t*-test was performed on both Hippocampus and Spleen datasets.

| Data Setting | Diff-CoGAN (without) Using Hippocampus Dataset | | | | Data Setting | Diff-CoGAN (without) Using Spleen Dataset | | | |
|---|---|---|---|---|---|---|---|---|---|
| | HD | | ASD | | | HD | | ASD | |
| | G1 | G2 | G1 | G2 | | G1 | G2 | G1 | G2 |
| 100–100 | <0.05 | >0.05 | <0.05 | >0.05 | 100–100 | <0.05 | <0.01 | >0.05 | <0.01 |
| 100–1000 | <0.05 | <0.05 | <0.05 | <0.05 | 100–400 | <0.05 | >0.05 | <0.05 | <0.01 |
| 100–2000 | <0.05 | <0.05 | <0.05 | <0.05 | 100–900 | <0.05 | <0.05 | <0.05 | <0.05 |
| 100–2900 | <0.01 | >0.05 | <0.01 | >0.05 | | | | | |

The extensive experiments demonstrated that the Diff-CoGAN framework can produce promising predicted maps. The results using two different datasets indicate the effectiveness and robustness of Diff-CoGAN. Although there are important discoveries revealed by these experiments, there are also limitations. Firstly, the Dice and IoU values were very close when more unlabeled data participated in the training of different models. At the same time, the HD and ASD values of Diff-CoGAN decreased steadily. This suggests that Diff-CoGAN can maintain segmentation accuracy and reduce boundary errors. Further study needs to be conducted to improve the significance difference in the Dice and IoU values between different models. Secondly, the networks of generators adopt different strategies, which are transfer learning [39] and Densenet [34]. The results of the model with transfer learning always performed better than those with Densnet (Refer Tables 3–10). This means transfer learning can more effectively extract information from the features maps due to the prior knowledge. While the two generators are both expected to be effective on labeled data and unlabeled data, finding a suitable network still needs to be investigated. Finally, Diff-CoGAN limits the image views to two views in this work. Future work should consider additional views and more generators to improve segmentation.

Overall, our results show that the adversarial training, the mutual information guidance of the two generators, and the intersection of the two predicted maps with high confidence region in the proposed Diff-CoGAN can effectively improve the segmentation accuracy for medical images. In the future, we will continue to focus on overcoming the limitations of the Diff-CoGAN framework.

## 5. Conclusions

Accurate segmentation of target areas from medical images is significant for clinical diagnostic procedures and quantitative analysis. Usually, automated segmentation of medical images depends on labeled data, which needs experts to provide detailed annotation. Large-scale unlabeled data contain a lot of information about target regions and deserve to be exploited in an automated segmentation system, especially when labeled data are scarce. In this paper, we propose a learning-based model using semi-supervised learning to extract information from unlabeled data to improve the accuracy of medical image segmentation.

Due to the complexity of medical images and imperfect dataset, it is a challenging task to improve target region segmentation accuracy with a small, labeled dataset and a relatively large-scale unlabeled dataset. In this paper, a semi-supervised learning method based on Diff-CoGAN was proposed to achieve medical image segmentation. The strategy of co-training and adversarial training of GAN were adopted to construct a collaborative segmentation framework named Diff-CoGAN. When Diff-CoGAN was compared to semi-GAN and co-training using the Hippocampus and Spleen datasets, the Dice and IoU values showed that Diff-CoGAN achieved the highest accuracy. Meanwhile, the HD and ASD metrics also proved that Diff-CoGAN produced better boundary segmentation. These results signified that the use of two generators and one discriminator in Diff-CoGAN was consistently effective in performing medical image segmentation. The segmentation performance of semi-GAN, which consists of one discriminator and one generator, and co-training, which comprises two generators, were inconsistent and no definitive conclusion could be drawn. Diff-CoGAN achieves higher segmentation accuracy through adversarial training because the discriminator can guide the two generators to generate more accurate predicted maps, and the mutual information guidance of the two generators can also promote the improvement of segmentation performance.

Our proposed Diff-CoGAN also introduces the intersection of two predicted maps, with the high-confidence region produced by the outputs of the two generators as the input to the discriminator. As shown earlier, the intersection operation managed to reduce the boundary error in segmentation, particularly when unlabeled data were increased. In addition, the generators in Diff-CoGAN adopted different deep learning networks in the encoder to increase the diversity of extracted features, consequently providing complementary information for the same data. Generator 1 used DenseNet and Generator 2 utilized VGG16 as their respective encoder.

Diff-CoGAN limits the image views to two views in this work. Future work should consider additional views to improve segmentation. The same Diff-CoGAN framework should also be further experimented on 3D medical images to investigate its robustness.

**Author Contributions:** Conceptualization, G.L. and N.J.; methodology, G.L.; software, G.L.; validation, G.L.; formal analysis, G.L., N.J. and R.H.; investigation, G.L.; resources, G.L.; data curation, G.L.; writing—original draft preparation, G.L.; writing—review and editing, N.J. and R.H.; visualization, G.L. and N.J.; supervision, N.J. and R.H.; project administration, N.J. and R.H.; funding acquisition, G.L., N.J. and R.H. All authors have read and agreed to the published version of the manuscript.

**Funding:** This research was funded by the Scientific and Technological Innovation Programs of Higher Education Institutions in Shanxi, grant number 2022L538, and the Youth Foundation of Taiyuan Institute of Technology, grant number 2018LG08.

**Data Availability Statement:** Not applicable.

**Conflicts of Interest:** The authors declare no conflict of interest.

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
