# Peer review of "An Improved Co-Training and Generative Adversarial Network (Diff-CoGAN) for Semi-Supervised Medical Image Segmentation"

_information, doi:10.3390/info14030190_

Round 1

Reviewer 1 Report

This article presents a semi-supervised learning method based on Diff-CoGAN to improve the medical image segmentation accuracy. However, the article is poorly written and lacks clarity, making it difficult for readers to follow the methodology and results. The abstract and introduction are particularly problematic, as they fail to provide a clear background and motivation for the proposed method.

The article also lacks critical evaluation of the proposed method. While the authors claim that Diff-CoGAN achieved higher segmentation accuracy compared to other methods, they did not provide any statistical analysis or comparison to state-of-the-art methods. The authors are invited to consult Abirami et al. (2021). P2P-COVID-GAN: Classification and segmentation of COVID-19 lung infections from CT images using GAN. International Journal of Data Warehousing and Mining, 17(4), 101-118; Dirvanauskas et al. (2019). HEMIGEN: Human embryo image generator based on generative adversarial networks. Sensors, 19(16). Moreover, the article fails to provide a comprehensive discussion on the limitations of the proposed method.

Although the experiments are quite extensive and many results are presented, the paper would benefit from the statistical analysis to compare the analyzed methods and statistical analysis to confirm or reject a hypothesis of equal mean. The article also would benefit from more details and explanations on the methodology and experimental setup. For instance, the article lacks information on how the labeled and unlabeled data were selected and pre-processed. The article would also benefit from more visualizations, such as sample images overlayed with ground truth and segmentation results, to help readers understand the proposed method and its performance.

Minor comments: Line 292: “to be rotated 180°” -> do the authors mean matrix transposition?

Overall, the article requires significant revisions to improve its clarity and to provide more in-depth analysis and evaluation of the proposed method.

Author Response

Dear Prof./Dr.

Thank you for the comments and recommendations. We really appreciate them. Please see the attachment for our responses to the comments. 

Reviewer 2 Report

This paper proposed a GAN-based approach to improve the performance of semi-supervised medical image segmentation. While results look promising, I have some doubts and hope authors to address:

1. The paper has some grammar errors and need to improve the writing. Please revise it. 

2. The figures are hard to read and understand. Please revise figures and provide more clear version. Also, please provide more explanation on the legends of figures 

3. The propose approach is very close to CycleGAN architecture since it contains 2 generators. Why CycleGAN is not used as benchmark? 

4. Dice is similar with IoU. Why dice and IoU are both used in this paper? What are their difference? 

5. What strategy authors have used to prevent overfitting during the training? Please provide more training and evaluation details. 

Author Response

(The authors gave the same response as above.)

Round 2

Reviewer 1 Report

The quality of the manuscript has been improved. The manuscript can be accepted for publication.

Reviewer 2 Report

The revision has improved the quality of this manuscript